# Improvements in blood and fitness tracker biomarkers in a longitudinal real-world cohort of digital health platform users

**Nimisha Schneider**📷*, **Paul Fabian, Michelle Cawley, Bartek Nogal, Gil Blander, Renee Deehan**

InsideTracker, One Broadway, Cambridge, Massachusetts, United States of America

* nschneider@insidetracker.com

## Abstract

Digital health technologies offer new opportunities for personalized health management and disease prevention. In this retrospective, long-term longitudinal study of over 20,000 users of a digital health platform (DHP), we aimed to determine whether improvements in health-related biomarkers could be observed in users and, if so, how they were maintained over time. We further explored whether genetic predisposition and physiological patterns, such as sleep and activity, were associated with variability in these biomarker responses. The DHP evaluates a user's individual biological profile, consisting of blood biomarkers, polygenic risk scores (PGS), and fitness tracker data, and provides personalized lifestyle interventions based on knowledge about nutrition, supplements, exercise, and recovery collected from over 7,000 clinical studies. Here, we show improvement in suboptimal levels of the primary outcome, key blood biomarkers that are sustained or increased in the long-term with DHP use. We additionally show the correlation of biomarker improvement with secondary outcomes, including specific sleep and activity patterns in users. Lastly, we find significant correlations between polygenic risk and both baseline levels and longitudinal change in biomarkers, including low-density lipoprotein cholesterol (LDL-c), suggesting that genetic predisposition for a negative trait (e.g., elevated LDL-c) could make it more difficult to improve that trait. This longitudinal, integrated biomarker dataset highlights the potential of digital health tools in fostering improvements in health-related biomarkers through personalized data analytics and targeted behavioral interventions.

## Author summary

Digital health applications promise to turn personal data into improved health and longevity, but large-scale, long-term evidence has been limited. We retrospectively analyzed multi-year data from more than 20,000 adults using a digital

**Data availability statement:** A de-identified minimal dataset sufficient to replicate the primary findings of this study is included as Supporting Information. Genetic data used in the study are not publicly available due to participant consent and data use restrictions. Access to the polygenic risk scores calculated from individual genetic data and used in this analysis may be requested from InsideTracker by contacting Slava Brodskiy via privacy@insidetracker.com for researchers who meet the criteria for access to confidential data.

**Funding:** This study was funded by InsideTracker. NS, PF, MC, and RD are employees of insidetracker and receive financial compensation from InsideTracker. BN was a salaried employee at InsideTracker during the investigative and writing phases. Thus, there is no distinction between the investigative team and funding source.

**Competing interests:** All authors are full time employees of InsideTracker and hold stock options but declare no non-financial competing interests. G.B. also holds InsideTracker shares. G.B. is a scientific advisory board member of the University of Illinois' Precision Nutrition Initiative and has received consulting fees from this institution. G.B. is also on the board of ScienceAbroad.

health platform that combines blood tests, wearable data (activity and sleep), and genetic information to give personalized lifestyle guidance about food, supplements, exercise, and recovery. We found that users who started with suboptimal blood biomarkers such as LDL cholesterol and HgbA1c showed meaningful improvements that were sustained across years and multiple follow-up tests. Data from wearables provided information on who improved: users who increased their daily steps by an average of ~1,000 steps from baseline to follow-up activity levels, and those with higher REM sleep percentages, were more likely to shift cholesterol in a healthier direction, despite substantial month-to-month variability in step counts. Genetics also characterized biomarker improvement: individuals with a higher inherited risk for certain traits (for example, high LDL) tended to show smaller improvements, suggesting a genetic influence on how much people benefit from lifestyle changes. Our results provide large-scale, real-world evidence that integrating blood, wearables, and genetics can guide practical, personalized actions and may help people maintain healthier biomarker levels over time.

## Introduction

The concept of extending healthspan (time spent living in good health) has gained significant traction in the field of healthcare, particularly as it pertains to non-medical interventions that can lower disease risk [1,2]. Lifestyle modifications, related to nutrition and exercise, can yield up to an 80% reduction in risk for the most common chronic diseases [3]. Digital health platforms are emerging as powerful tools for individuals, together with their medical teams, to actively monitor personal health [4]. These platforms provide a unique opportunity to longitudinally gather comprehensive health and behavioral data, facilitate personalized interventions, and track progress.

The impact of lifestyle change on health outcomes and disease prevention is well established. Large cohort studies including the National Health and Nutrition Examination Survey (NHANES) [5], the Framingham Heart Study [6], the Nurses Health Study [7], and the UK Biobank [8] provide insight on how lifestyle factors influence population health. Retrospective, observational studies from these cohorts have indicated the importance of physical activity and diet to cardiovascular, metabolic, and even mental health outcomes, as well as to overall longevity [9–12]. These large-scale efforts have driven policy decisions, shaping global nutrition and exercise guidelines, and refining clinical biomarker reference ranges [13]. Smaller cohorts from private health companies have also leveraged consumer data to provide real-world insights about population-level health. For instance, the metabolic health company Zoe reported that extra weekend sleep negatively impacts the gut microbiome [14]. Arivale, a personalized health coaching company, demonstrated that genetics can influence how clinical biomarkers improve with lifestyle interventions [15].

Other studies have indicated that receiving education and/or coaching about recommended healthy lifestyle interventions is an effective health improvement

strategy. A study of middle-aged women found that those who received health education and lifestyle coaching were able to improve their cardiovascular biomarker values at follow-up [16]. Similarly, a lifestyle intervention program targeting metabolic syndrome showed positive outcomes associated with progression to diabetes in high-risk individuals [17]. Digital health platforms (DHPs) have emerged as scalable and accessible mechanisms for delivering such lifestyle education and coaching, particularly for populations with limited access to traditional healthcare. A recent meta-analysis of digital interventions for prediabetes demonstrated that DHPs including apps, virtual coaching, and web-based platforms can lead to improvements in glycemic control, physical activity, and weight management, highlighting their potential to effectively support behavior change at scale [18].

While some of the above cohorts have collected data from wearable devices (fitness trackers), often it was for a short period of time (1–2 weeks) and relied on early generations of fitness trackers that did not accurately capture activity, sleep data, or advanced metrics like vO2 max estimates. Wearable device data has been shown to correlate with cardiovascular symptom and disease data, as well as user-reported quality of life [19]. This evidence supports wearables' utility in continuous health monitoring and healthspan assessment. However, few studies have reported comprehensive systematic surveys of clinical blood parameters and commonly worn wearables metrics to date, which provide real-world evidence of the applicability of such devices.

Here, we examine the use of a DHP that provides personalized, evidence-based nutrition, supplement, exercise, and recovery interventions based on integrated blood, DNA, and wearables-based personal biomarker data. The DHP studied is a wellness-focused tool intended to support health optimization through lifestyle insights and is not intended for medical device certification under current regulatory guidelines (see Methods). Our primary objective was to determine whether improvements in suboptimal blood biomarker levels could be observed in DHP users and whether such improvements were sustained over time. Secondary objectives were to examine whether biomarker response varied by physiological patterns captured via fitness tracker data (e.g., sleep, activity), and to assess whether genetic risk influenced the degree of improvement. We reported on an analysis of 38 clinical blood biomarkers from a longitudinal, retrospective consumer cohort of 1032 individuals in 2018 [20]. Here, we expand our investigation to >20,000 individuals and 39 clinical blood biomarkers that are relevant in predicting age-related diseases and assessing overall health [21–25]. Two additional data modalities are also incorporated: 1) physiological performance measurements captured passively from on-market fitness trackers, including estimated vO2 max, resting heart rate (RHR), sleep architecture (e.g., REM sleep %), and activity characteristics; and 2) genetic predispositions for measured biomarkers levels. We replicated the average biomarker improvement correlated with DHP use from the initial cohort and demonstrated sustained improvement over a longer time frame, as well as identified new associations between blood and genetic risk. We further distinguish the fitness tracker data characteristics of users able to improve biomarker values from those that are not.

These results hold substantial implications for individuals seeking to proactively manage their health through personalized, evidence-based lifestyle "prescriptions". Incorporating fitness tracker metrics can provide daily, directional feedback to individuals about their health, and the genomic information can inform when lifestyle interventions have reached their potential. Our findings contribute to the growing body of knowledge that bridges digital innovation and health enhancement, spotlighting the potential of technology to reshape our approach to preventive health.

## Results

### Cohort selection and characteristics

The baseline cohort of 20,342 subjects with at least two blood tests measuring up to 39 biomarkers were included in this study (S1 Table). Subjects could add DNA and/or wearable fitness tracker data to their blood biomarker profiles (S2 Table). Follow-up analyses were done at least 90 days post-baseline (median interval = 260 days). Post-baseline, users were recommended to engage in personalized fitness, lifestyle, supplement, and nutrition interventions tailored to their biomarker results and survey input.

 

PLOS Digital Health

Users (Table 1) were predominantly males (64.2%) and white (84.3%). Males and females exhibited similar break-downs in ethnicity and mean age (~46 years). Mean Body Mass Index (BMI) for females (23.2 kg/m$^2$) was within the World Health Organization's parameters for normal weight, while mean BMI for males was slightly higher (25.3 kg/m$^2$), placing them just over the limit into the "pre-obesity" category [26]. Disease diagnoses, symptoms, or treatments were not collected as part of the digital health platform.

## Genetic risk and baseline blood biomarker values

To establish how genetic risk correlates with cohort biomarker levels, we performed linear regressions between PGSs and the respective baseline blood biomarker levels in genotyped users with blood data (~38% of cohort). Table 2 summarizes PGSs that explain more than 5% of the variance within blood biomarker levels, suggesting that hematological and lipid traits have a relatively stronger genetic component relative to other blood biomarkers. (See S5 Table for full list of PGS-biomarker correlations).

**Table 1. Digital Health Platform User Cohort, Demographics.**

|  | Female | Male |
|---|---|---|
| Users | 7212 | 13130 |
| Age (mean (SD)) | 47.20 (11.77) | 46.65 (11.88) |
| Body Mass Index (mean (SD)) | 23.21 (3.77) | 25.29 (3.29) |
| **Ethnicity (%)** |  |  |
| Asian | 460 (6.4) | 637 (4.9) |
| Black | 155 (2.1) | 285 (2.2) |
| Hispanic | 415 (5.8) | 715 (5.5) |
| Native American | 15 (0.2) | 22 (0.2) |
| South Asian | 110 (1.5) | 389 (3.0) |
| White | 6057 (84.0) | 11064 (84.4) |

SD: standard deviation.

**Table 2. Correlation between polygenic traits and baseline blood biomarker levels.**

| Polygenic risk score trait | Baseline blood biomarker phenotype | % phenotype explained by trait | P-value |
|---|---|---|---|
| mean platelet volume | mean platelet volume | 12.84% | <0.005 |
| total iron binding capacity | total iron binding capacity | 10.86% | <0.005 |
| ApoB | apolipoprotein B | 10.03% | <0.005 |
| mean corpuscular hemoglobin | mean corpuscular hemoglobin | 9.50% | <0.005 |
| platelet count | platelets | 8.29% | <0.005 |
| mean corpuscular volume | mean corpuscular volume | 6.95% | <0.005 |
| low density lipoprotein cholesterol | low density lipoprotein cholesterol | 5.88% | <0.005 |
| high density lipoprotein cholesterol | high density lipoprotein cholesterol | 5.67% | <0.005 |
| total cholesterol | total cholesterol | 5.03% | <0.005 |

## Biomarker improvements are observed *between* first and second blood draws

Biomarker values are assigned a "zone": if within the clinically denoted reference range, it is considered "normal", but if above or below that zone, is marked as "out of range". Within the normal zone, the biomarker can also fall into an "optimal" zone, based on age, sex and prior knowledge (see Methods). To determine whether users with a non-optimal biomarker value can improve, we analyzed those with two or more values that spanned at least a 90-day interval. Changes in mean levels were assessed from baseline for all biomarkers, with users with above-optimal baseline levels analyzed separately from those with below-optimal levels (Table 3 and Fig 1; see S3, S4 Tables for full list). All biomarker changes displayed in Fig 1 were statistically significant by Mann-Whitney U test ($p < 0.05$). Due to our large sample size, many of the markers showed statistically significant improvement upon retest in users with initially suboptimal levels; we thus present effect size information (median (IQR) at each draw as well as percent of users who improved enough to enter a new zone or achieved optimal levels).

Some biomarkers exhibited significant shifts towards optimal levels upon retesting, but the degree of change varied. Improvement was defined as a shift in levels that moved users into a zone closer to optimal. A higher proportion of users improved their vitamin B12, D and folate (Fol) levels (36.1%, 57.1%, and 63.2%, respectively, Fig 1G, 1H, 1I) compared to LDL-c and high-density lipoprotein cholesterol (HDL-c) (20.4% and 27.3% respectively, Fig 1A, 1E).

A high proportion of users were able to improve HgbA1c, fasting glucose, triglycerides, and hsCRP levels (79.3%, 74.2%, 76.2%, and 70.7% respectively; Fig 1E, 1C, 1B, 1F), but relatively few could completely optimize them (21.4%, 31.1%, 29.6%, and 25.4%, respectively, of suboptimal users improved). LDL-c also showed this pattern: a relatively higher proportion (20.4%) of users able to improve vs (5.3%) optimize their initially high levels by their follow-up blood test.

**Table 3. Users with non-optimal baseline biomarker levels: baseline vs. follow-up.**

| Biomarker | Cohort | Draw 1 median(IQR) | Draw 2 median(IQR) | Unit | % users improved* | % users optimized* | Paired samples |
|---|---|---|---|---|---|---|---|
| cortisol | high baseline level | **18.9(4.2)** | **15.6(6.9)** | µg/dL | 73.7% | 41.0% | 3027 |
| fasting glucose | high baseline level | **95(8)** | **91(11)** | mg/dL | 74.2% | 31.1% | 9180 |
| triglycerides | high baseline level | **113(48)** | **99(57)** | mg/dL | 76.2% | 29.6% | 7215 |
| hsCRP (C-reactive protein, high sensitivity test) | high baseline level | **1.5(1.7)** | **1.2(1.7)** | mg/L | 70.7% | 25.4% | 5005 |
| HgbA1c (glycated hemoglobin) | high baseline level | **5.4(0.3)** | **5.4(0.3)** | % | 79.3% | 21.4% | 5533 |
| LDL-c (low-density lipoprotein cholesterol) | high baseline level | **149(27)** | **141(37)** | mg/dL | 20.4% | 5.3% | 5128 |
| folate | low baseline level | **4.941(0.3)** | **8.9665(6.4)** | ng/mL | 63.2% | 62.7% | 204 |
| vitamin D | low baseline level | **26(7)** | **35(17)** | ng/mL | 57.1% | 51.7% | 5138 |
| iron | low baseline level | **61(21)** | **92(46)** | ug/dL | 63.9% | 35.6% | 1574 |
| vitamin B12 | low baseline level | **377(130)** | **437(200)** | pg/mL | 36.1% | 27.9% | 3423 |
| testosterone (males only) | low baseline level | **415.25(120)** | **452(180)** | ng/dL | 30.0% | 25.5% | 3720 |
| HDL-c (high-density lipoprotein cholesterol | low baseline level | **49(11)** | **51(15)** | mg/dL | 27.3% | 20.2% | 7977 |

IQR: Inter-quartile range.

Bolded values indicate $p < 0.05$. See S3 Table for p-values.

* "Improved" refers to users who decreased higher/raised lower than optimal levels by at least one zone; "Optimized" refers to users whose levels were in the optimal zone at draw 2 (see Methods for additional detail).

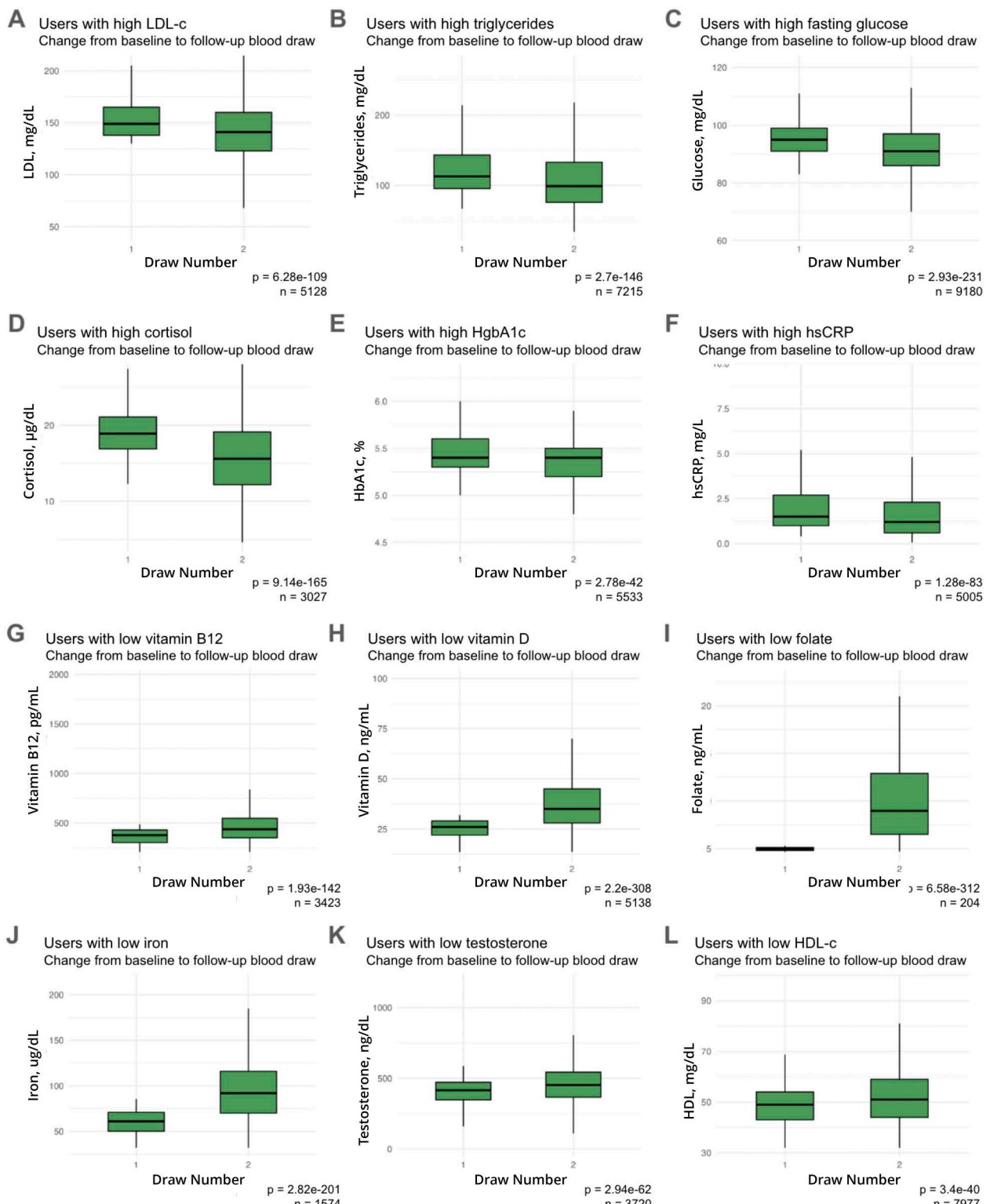

**Fig 1. Overall improvement at subsequent blood tests in non-optimal biomarkers at baseline.** Biomarker levels for users are shown at baseline and at follow-up blood test. A, LDL; B, Triglycerides; C, Fasted Glucose; D, Cortisol; E, HgbA1c; F, hsCRP; G, Vitamin B12; H, Vitamin D; I, Folate; J, Iron; K, Testosterone (males only); L, HDL Cholesterol. Mann-Whitney U-test p-values and user paired sample size are shown.

## Users *with* multiple draws exhibit sustained improvements *in* biomarker profiles

To explore the long-term sustainability of improvements in biomarker levels, we examined a subset of users who underwent more than two blood tests, conducted 90+ days apart, receiving updated lifestyle interventions after each test. Our objective was to ascertain whether improvements in biomarkers, initially suboptimal at baseline, were maintained over 5 or more blood draws (average elapsed time 4.2 yrs, Fig 2). On average, users maintained - and in some cases further improved - their biomarker levels over this extended period. Initial improvements in fasting glucose, cortisol, HbA1c, hsCRP, and folate levels were sustained (Fig 2C, 2D, 2E, 2F, 2I). Triglycerides, vitamins B12 and D, iron, testosterone (in males), and in LDL-c and HDL-c levels, showed continued improvements after the second test (Fig 2B, 2G, 2H, 2J, 2K, 2A, 2L).

To examine the direct clinical, disease-related impact of longitudinal biomarker improvements, we focused on HgbA1c, a diagnostic tool for Type 2 diabetes (T2D) [27], and examined a small subgroup of users (n = 24) with baseline HgbA1c of 6.5% or higher (cutoff for diagnosing T2D). HgbA1c levels decreased over five longitudinal tests, with the draw 5 average achieving pre-diabetic levels (6.28%, Fig 3).

## Characteristics *of* users who improve baseline biomarker levels upon retest

**Genetic risk correlates with specific biomarker-level changes.** To examine the impact of genetic predispositions on biomarker fluctuations, we categorized users' PGS results into tertiles (labeled T1, T2, T3; defined as lowest 10%, middle 80%, and highest 10% of scores) and correlated these with longitudinal changes in biomarker levels in our cohort. The ferritin PGS most significantly associated with longitudinal changes with the corresponding serum biomarker (Fig 4C): high baseline ferritin levels decreased on average upon retest in T1, T2, and T3 users; however, the decrease was significantly larger for users in T1 than those in T2 or T3. A similar but smaller magnitude correspondence was seen between both LDL-c and total cholesterol (TC) PGS and longitudinal LDL-c and TC blood level changes (Fig 4A, 4B). No other PGS examined exhibited a statistically significant effect (S6 Table).

**Higher activity levels between baseline and follow-up tests correlate with specific biomarker-level improvement.** To assess whether activity or sleep differences are associated with users who show biomarker improvements compared to those who do not, we examined a small cohort of individuals (n = 126) with high TC levels at baseline, and who had connected a wearable device to the platform. This cohort was divided into users who improved (changed their biomarker levels by at least one zone towards the "optimal" zone, or lowered their biomarker levels by 10% or more, regardless of zone movement) and those who did not improve. Users who improved TC from baseline had similar baseline fitness and activity levels to those who did not (S7 Table). However, those who improved showed a significant increase in activity, exemplified by step count, between the baseline and follow-up tests, while users who did not improve maintained their activity levels. Users who improved increased their baseline average step count of 8700 steps by an average of 950 steps per day, compared to users who did not improve, who did not increase their average step count (Fig 5A). Additionally, both at baseline and throughout the time between blood tests, users who improved TC had significantly higher REM sleep percentages (22% on average for improvers vs 18% for non-improvers, Fig 5B).

## Discussion

In this study, we investigated associations between engagement with a digital health platform and longitudinal changes in blood biomarkers in a large cohort of generally healthy adults, using data from blood biomarkers, genotyping, and wearable devices. We aimed to evaluate whether personalized, evidence-based nutrition, supplement, exercise, and recovery interventions in the form of customized recommendations based on integrated blood, DNA, and wearables-based personal biomarker data were associated with improvements in key blood biomarkers among users of a digital health platform. We also aimed to explore how differences in physiological markers (e.g., sleep and activity patterns) and genetic

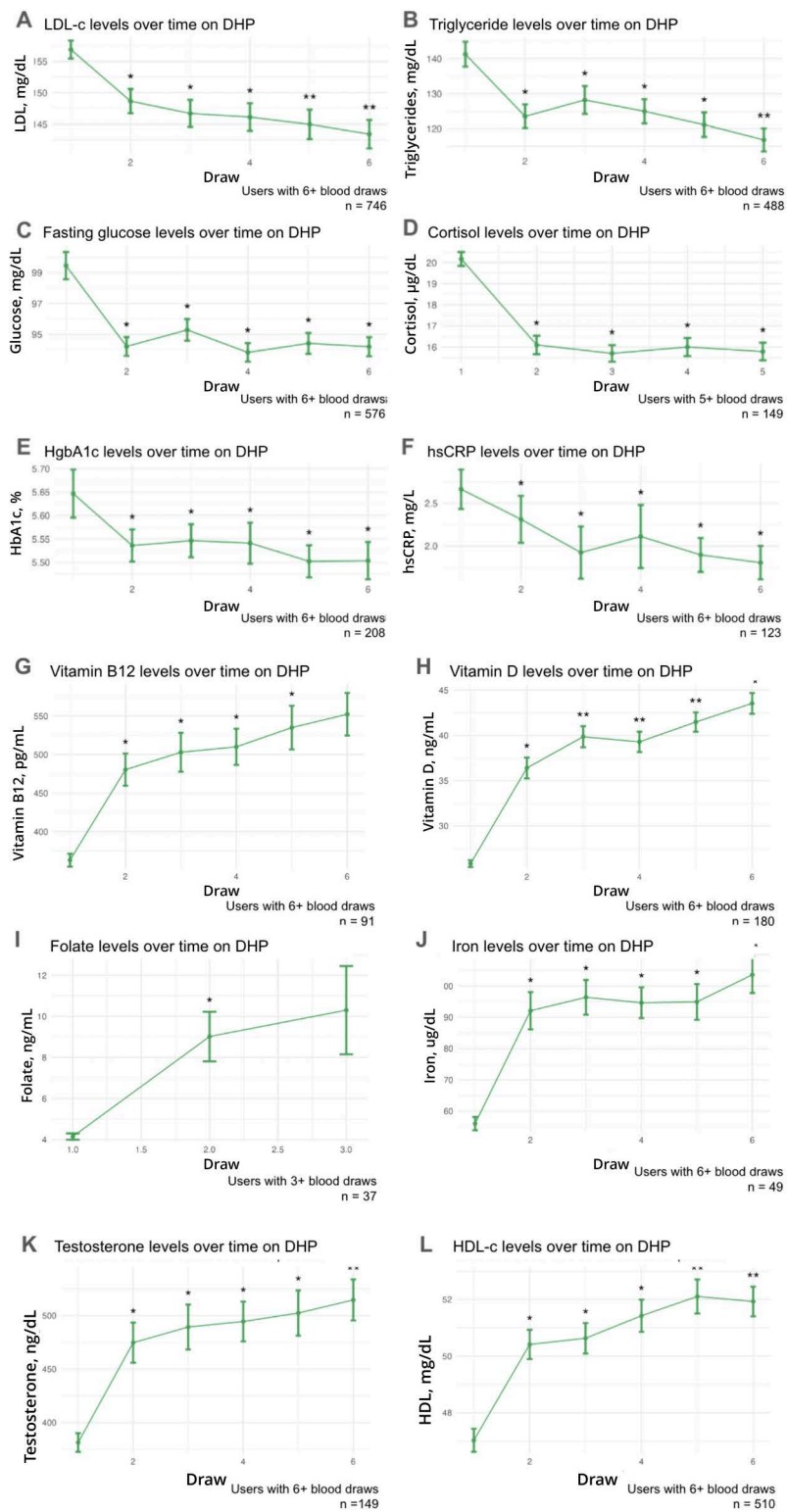

**Fig 2. Long-term user trajectory of improvement in non-optimal at baseline biomarkers.** Biomarker levels for users are shown at baseline and at follow-up blood tests. A, LDL; B, Triglycerides; C, Fasted Glucose; D, Cortisol; E HgbA1c; F, hsCRP; G, Vitamin B12; H, Vitamin D; I, Folate; J, Iron;

K, Testosterone; L, HDL Cholesterol. A single asterisk indicates the mean value at that draw is significantly different from that of draw 1; two asterisks indicate the mean value at that draw is both significantly different from that of draw 1 and of the first follow-up draw (draw 2). Standard error bars and sample size are shown. User samples are matched throughout draws shown.

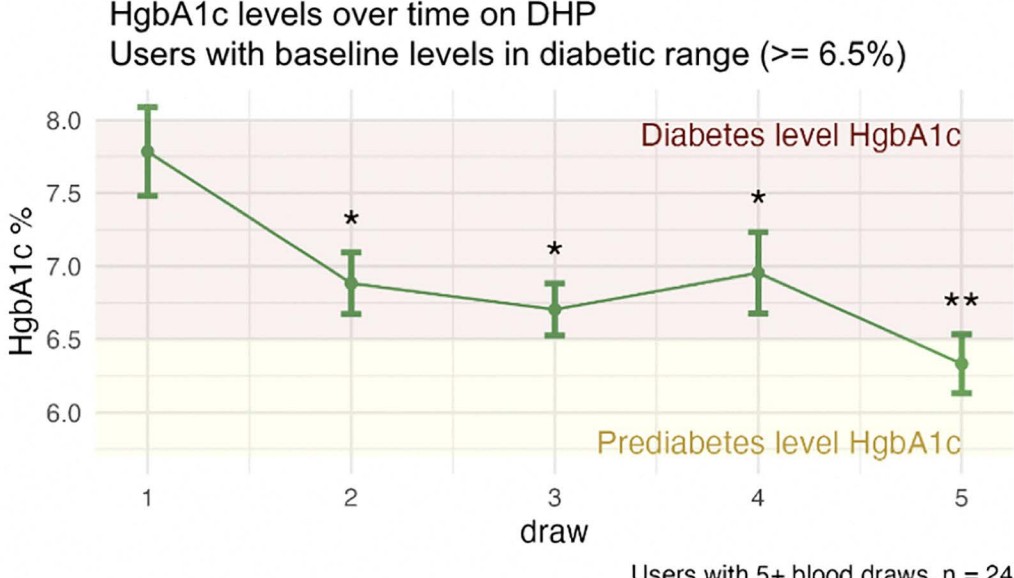

**Fig 3. Long-term user trajectory of improvement in clinically high HgbA1c at baseline.** A single asterisk indicates the mean value at that draw is significantly different from that of draw 1; two asterisks indicate the mean value at that draw is both significantly different from that of draw 1 and of the first follow-up draw (draw 2). Standard error bars and sample size are shown. User samples are matched throughout draws shown.

risk were associated with variations in biomarker change over time. Our findings suggest that prolonged engagement with such a platform correlates with improvements in blood biomarker levels over several years and multiple follow-up tests. Furthermore, we identify that genetics and activity levels can distinguish users who can improve versus those who do not.

## Genetic predisposition and biomarker changes

In users with baseline high levels of ferritin, LDL-c, and TC, those with a low genetic propensity for high levels of these biomarkers were able to significantly lower their levels of those markers at follow-up compared to those with a high genetic predisposition (Fig 4). This result aligns with and adds to previous work that reported a similar interaction between higher genetic risk of LDL-c, HDL-c, triglycerides, and TC and the ability to modify the respective biomarkers' serum levels [15]. This underscores the significance of genetic information for individuals unable to make sufficient changes in biomarkers through lifestyle changes alone.

## Longitudinal improvement

Many of the markers examined showed statistically significant improvement upon retest of users with unoptimized levels, though the magnitude of this improvement varied by biomarkers. Some variation may be explained by the type of intervention recommended to improve a specific biomarker. For example, suboptimal vitamin levels can be addressed by supplementation, a relatively easy lifestyle intervention.

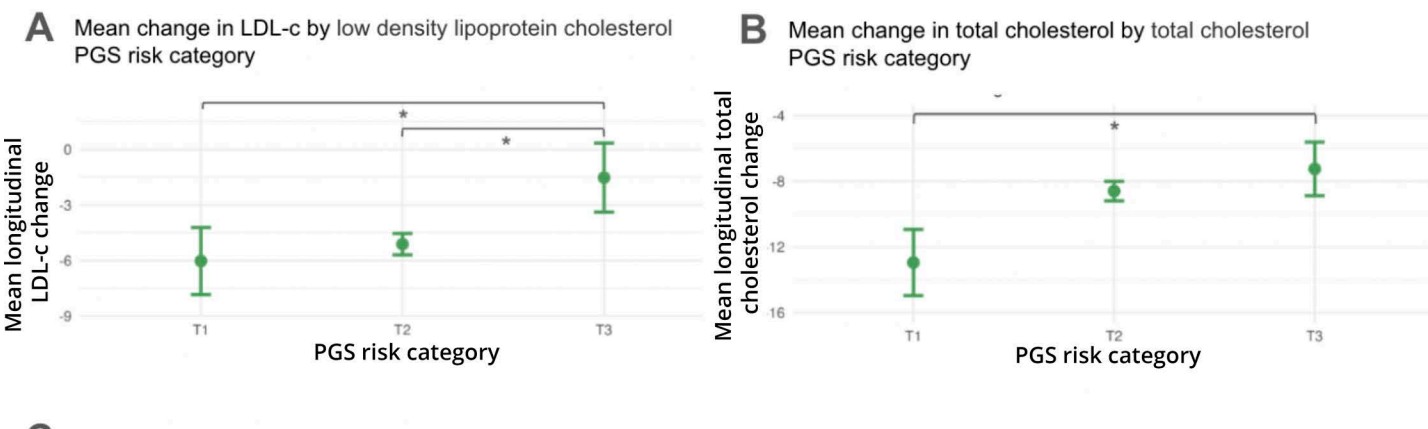

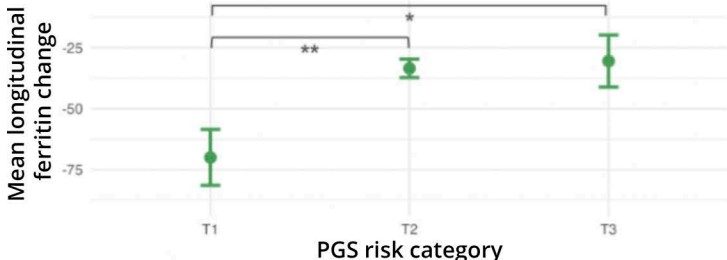

**Fig 4. Genetic traits calculated based on polygenic score correlate significantly with longitudinal change in biomarker levels.** A, LDL trait calculated based on PGS correlates significantly with longitudinal change in blood LDL levels. B, Total cholesterol trait calculated based on PGS correlates significantly with longitudinal change in blood cholesterol levels. C, Ferritin trait calculated based on PGS correlates significantly with longitudinal change in blood Ferritin levels. Mean change in above normal LDL, Cholesterol, and Ferritin levels from baseline to follow-up blood test (at least 90 days subsequent) is shown by corresponding genetic risk score category (T1 = lowest risk, T3 = highest risk). **t-test p-value <.005; *t-test p-value < 0.05.

However, less than 25% of users were able to optimize their high HgbA1c, TC, LDL-c, and APOB, as well as their low HDL. Interventions suggested to address these biomarkers are typically related to nutrition, exercise, and sleep, habitual behaviors which may be more challenging than supplementation for users to implement [28]. Additionally, lipid and glycemic markers likely take longer to modify with lifestyle interventions than our minimum time frame of 90 days, particularly for individuals with higher genetically determined risk (e.g., for LDL-c and TC, discussed above). Other studies that evaluated dietary interventions for high TC and HgbA1c suggest that, while changes can be seen in 6 weeks in a clinical setting, changes in observational, community-based studies are typically of lower magnitude and take a period of at least several months [29,30].

We observed significant, continued improvement after the first follow-up blood draw in users with initially suboptimal LDL-c and HDL-c, triglycerides, vitamins B12 and D, iron, and in males, free testosterone. For fasting glucose, cortisol, hsCRP, folate, or HgbA1c, participants with initially high levels significantly improved and then maintained the improved levels over multiple years. Across biomarkers, on average, the largest improvement was observed from baseline to the first follow-up draw. Since we selected our baseline user cohort specifically by those with out-of-range values, this shift in average levels upon follow-up may reflect a statistical regression to the mean (RTM) due to our non-random selection of baseline values [31]. Because these baseline values are subject to random variation (e.g., due to technical variability in test results), the chosen subset of participants would be expected to show

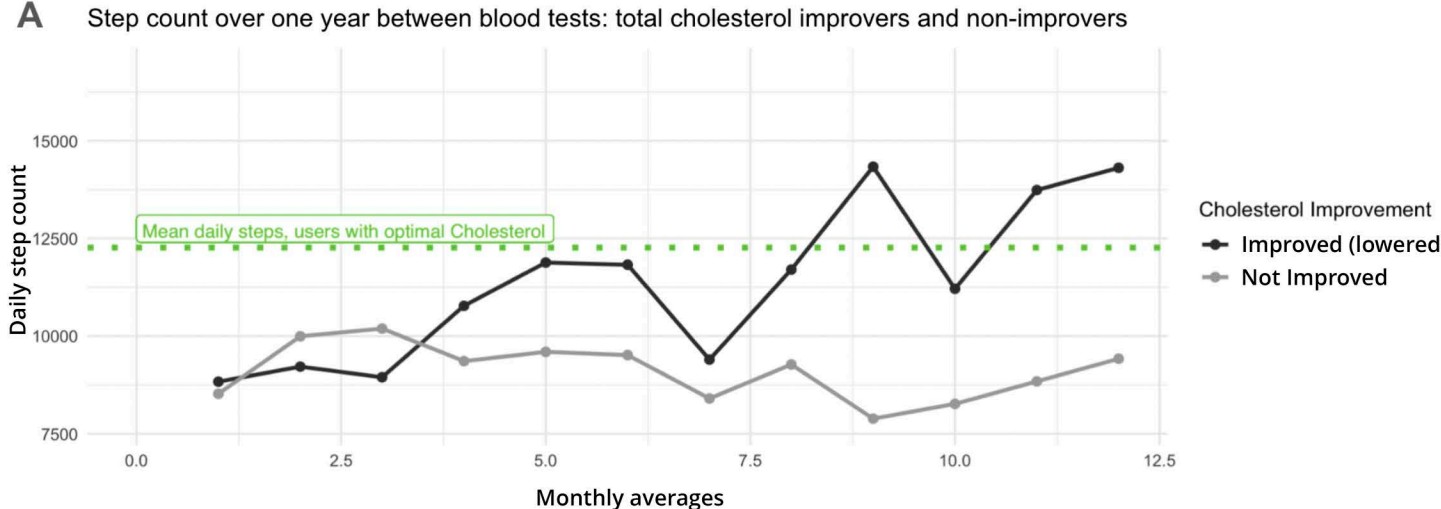

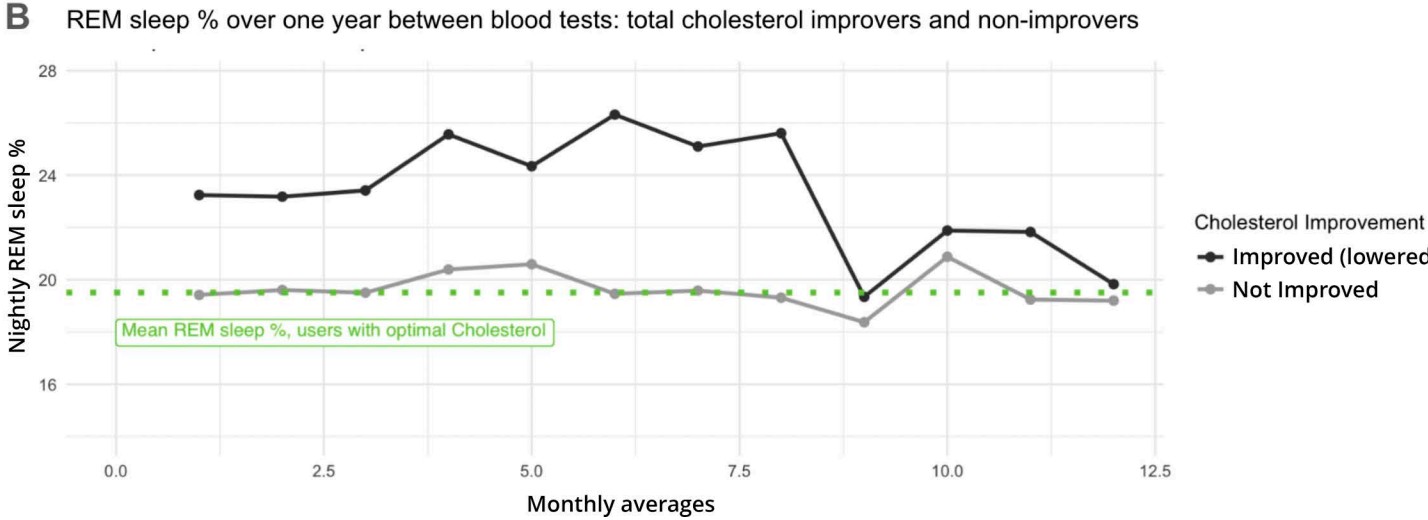

**Fig 5. Higher step count and REM sleep characterize a user subgroup with high total cholesterol at baseline, which improves at follow-up test.** A, Step count and B, REM sleep % are shown as a monthly average for 12 months post-baseline blood draw, for users who had high Cholesterol at baseline and went on to improve or not improve their Cholesterol levels by follow-up blood test.

improved follow-up test values simply by chance. Though this may also explain some of the observed improvement in Fig 2, particularly from baseline to second draw, the longitudinal stability of the initial change makes it unlikely that this effect is the primary explanatory factor. Estimated RTM effects varied substantially by biomarker when quantified explicitly (S10 Table), with higher estimated contributions for markers known to exhibit greater short-term biological variability (e.g., Cortisol, hsCRP). However, across biomarkers, initial improvements (where RTM would have maximum effect) were maintained or further improved across subsequent measurements. The larger improvement in biomarker levels observed from baseline to second draw in multiple biomarkers (Fig 2) may be attributed to some statistical artifact, but also to the notion that initial lifestyle changes cause a specific shift in biomarker levels, which, for many users, can be maintained and gradually increased, but not necessarily continued at the initial rate of change.

## Characteristics *of* improvers

Users with initially high TC levels are recommended to increase physical activity and receive real-time support and life-style advice related to their individual wearable fitness tracker data, if available. Users with initially high TC levels who subsequently lowered them also significantly increased their step count relative to those that did not improve. Further, we observed that the new step count of users who saw TC improvement was similar to the average step count of those with initially optimal TC levels (Fig 5), supporting the idea that higher activity levels are needed for optimal lipid profiles. While many studies have emphasized the importance of physical activity to cardiovascular health [32], our direct observation of step count correlation to change in a key cardiovascular health marker is an important finding for users seeking to maximize the impact of their lifestyle changes on their health.

We observed a higher average nightly REM sleep percentage in users who improved their TC levels, both initially (at the time of their baseline blood draw) and throughout the time between draws. Other studies have suggested decreased REM sleep as a cardiovascular risk factor [33]; our findings are interesting in that we do not observe changes in REM sleep associated with changes in TC, as with activity levels, but rather that users with more REM sleep at baseline find it easier to shift their initially high TC levels. It may be that users who get better quality sleep find increasing activity easier. Several studies have shown that increasing exercise subsequently improves sleep quality [34]; the impact of good sleep on propensity for activity is less studied, although certain studies have indicated it as a positive factor [35]. As our cohort of users with fitness tracker data grows, we hope to examine changes in additional biomarkers and their potential connections to physiological performance in a similar manner, and expand our understanding of the correlations we see between sleep and activity and cardiovascular health.

These wearable-associated analyses characterizing TC improvers should be considered exploratory. The subgroup of users with available fitness tracker and sleep data was relatively small (n = 126), and a relaxed significance threshold ($\alpha = 0.1$) was applied to balance limited statistical power against the substantial day-to-day variability of wearable-derived measures. As such, these findings are best interpreted as preliminary signals that motivate further investigation, rather than as definitive evidence, and will be reassessed as larger wearable-enabled cohorts become available.

## Limitations

This is a real-world cohort of United States-residing DHP users with biases including an overrepresentation of individuals identifying as male and white, and possessing the means to purchase our product and have access to technology. Users are likely health-conscious and have the drive and means to modify their behaviors. Lack of information about medications or medical conditions hinders comprehensive health and outcomes assessments. Our dataset pools sleep, fitness, and activity data from various wearable devices; addressing and normalizing device-specific variance [36] is an important avenue for future analyses.

Our retrospective, observational design, lacking a control cohort, precludes causal inferences regarding relationships between user behaviors, biomarkers, or among biomarkers themselves. This was an exploratory analysis, and conclusions are hypothesis-generating and correlative in nature. Ideally, a prospective interventional follow-on study in a less biased cohort would include: 1) a control arm where users were shown their results but did not receive recommendations, 2) required tracking of adherence in the intervention arm.

To partially address the lack of a control group and to inform the causal hypothesis, a negative-exposure comparator was examined: users with optimal baseline levels who therefore received no biomarker-targeted guidance. Across several key markers, this group shows a modest and slow drift away from optimal biomarker levels over time (S3 Fig), whereas users starting at suboptimal levels (and receiving DHP intervention) show sustained improvements (Fig 2). This comparator is not a counterfactual—it differs at baseline and remains subject to confounding—but it provides natural-history

biomarker trajectory context and supports the hypothesis that targeted recommendations contribute to the improvements observed in suboptimal users. Cross-sectional age trajectories from baseline tests (S4 Fig) also show gradual movement toward less favorable biomarker levels with increasing age. Consistent with this view, historical population data show modest upward trends in biomarkers such as HgbA1c over time in the general U.S. population [37], reinforcing that the improvements seen here are unlikely to reflect background trends alone. To address the issue of causality in the future, techniques like 2-sample Mendelian randomization could be applied by developing traits linked to compliance behaviors in our cohort, as sample size allows.

Adherence to recommendations is self-reported but doing so is not required; as such, the data is sparse and was not taken into consideration in this study. To support for the notion that at least partial compliance with recommendations is widespread in this user cohort, user engagement with the platform's Action Plan (AP) feature (selection of biomarker-targeted recommendations with in-app reminders) and its association with greater improvement at first follow-up is presented in S8–S9 Table (zone-shift tests with multiple-testing control), but this should not be interpreted as causal. Notably, AP uptake was high; 72% of users with 2 or more blood tests created at least one AP in the baseline to follow-up test window, indicating broad engagement with the behavioral component of the platform. While not a direct adherence measure, this level of engagement is consistent with relatively high compliance in the cohort (S2 Fig). Additionally, a small user follow-up survey (n = 53) suggested high self-reported adherence (~85%), yet selection and recall biases limit inference. Prior literature supports the idea that information/education alone can improve cardiometabolic biomarkers, consistent with a compliance-mediated mechanism. For example, dietitian-led nutrition education in dyslipidemia has been associated with reductions in total cholesterol, LDL-C, triglycerides, and HbA1c [38], and providing written lifestyle guidance to adults with type 2 diabetes has been linked to greater improvements in fasting glucose and lipid levels [39]. Prospectively, we plan to collect structured adherence signals to enable dose–response modeling and stronger causal analyses.

Because proxy variants for missing SNPs were derived using linkage disequilibrium patterns from a European-ancestry reference population (CEU), polygenic risk score estimates may be less accurate for users of non-European ancestry. As a result, associations involving PGS should be interpreted with caution in these groups and may underestimate true effects.

## Cohort diversity

While the current study provides valuable insights, its generalizability may be limited due to the predominantly White study population, which may not capture key biological and social factors affecting other racial and ethnic groups. To properly address both the biological and social determinants of health that are specific to non-White ethnic groups, we believe a dedicated follow-up study is necessary to examine health outcomes in Black, Latino/Hispanic, and Indigenous populations. Currently, we have longitudinal data from 440 Black users (155 female, 285 male), 1,130 Hispanic/Latino users (415 female, 715 male), and only 37 Indigenous users (15 female, 22 male), the latter of which will likely require specific targeted recruitment efforts to enable meaningful analysis. The importance of this work is underscored by well-documented examples such as the paradoxical observation that Black Americans often have favorable lipid profiles (e.g., low triglycerides and high HDL) despite elevated risk for cardiovascular disease [40], highlighting the limitations of applying standard biomarkers without consideration of population-specific contexts. Similarly, Indigenous populations face challenges with the accuracy of conventional metabolic disease diagnostics, where typical markers may fail to reflect true risk [41]. Complicating these issues is the distinction between race—often a proxy for social determinants of health—and ancestry [42], which may more directly capture biological variation. For instance, our current ethnic self-identification framework groups Latino and Hispanic individuals under a single identifier and encompasses broad ancestral diversity within the "Black" category. Where feasible, incorporating DNA-based measures of ancestry may help clarify biologically relevant differences while continuing to acknowledge the critical role of social determinants captured by self-identified race and ethnicity.

Recommendations require evidence of benefit from multiple independent clinical studies, and are grounded in universally accepted health principles, e.g., balanced nutrition, regular physical activity and adequate rest, that form the foundation of preventive health for most individuals. Ideally, digital health solutions of this kind would be integrated into government-funded universal healthcare programs, where some of all of the biomarkers are already part of routine testing, thereby substantially reducing the cost burden for users. To enhance accessibility in areas like the Global South, the DHP could integrate biomarkers relevant to local health priorities and ensure functionality in low-connectivity environments. Recommendations can be culturally adapted and available in local languages, and aligned with region-specific dietary practices and resource availability. Such measures would help bridge technological, financial and cultural barriers, enabling broader population benefit in low-resource settings.

## Impact

The integration of blood biomarkers, physiological performance data, polygenic risk scores, and user engagement in health interventions from a digital health platform presents a uniquely comprehensive model for understanding and improving healthspan. The scale, multimodal integration, and multi-year follow-up distinguish this work from prior reports and move the literature beyond marketing claims toward peer-reviewed evidence of DHP-associated health change. To our knowledge, this is the first study to show that platform-correlated improvements in clinically accepted biomarkers (e.g., LDL-C, HbA1c, selected micronutrients) are sustained across years and multiple retests. We further show that genetic risk relates not only to baseline phenotypes but also to subsequent change for specific markers (e.g., higher lipid-risk PGS associates with smaller LDL improvements). Finally, simple, interpretable behavioral signals from wearables, higher REM sleep percentage and an average of approximately 1,000 additional daily steps between tests, characterize users who shift cholesterol profiles in a healthier direction, offering practical targets observed in a real-world cohort.

While the DHP described in this study is positioned as a wellness tool and does not offer diagnostic or therapeutic services, we recognize the importance of aligning its use with clinical oversight—particularly when users receive results that suggest elevated clinical risk. We encourage integration of DHP outputs into conversations between users and their healthcare providers, explicitly instructing any user receiving a biomarker result that indicates emergent clinical risk to consult a physician. Given the clinical relevance of the biomarkers tracked in this product, usage of the DHP could be coupled together as part of routine care, under the guidance of a health professional in order to couple non-prescription to formal medical management. As the field of digital health continues to grow, the insights garnered from this research can inform the development of more targeted and effective strategies for health optimization and aging well.

## Methods

### Dataset

InsideTracker (insidetracker.com) is a for profit company that sells access to a personalized DHP that uses data from clinical blood tests with options to add genomics and fitness tracker information to generate evidence-based lifestyle recommendations. Recommendations are developed by evaluating published research around a particular question, e.g., what is the impact of daily oat consumption on LDL-c levels, and if sufficient evidence from multiple clinical studies was found, it is parsed to the user algorithmically through logic encoded in the expert system, e.g., a user may only see a recommendation to consume oats if their LDL-c is above optimal, a vegetarian user would be given a recommendation to consume meat, etc. (for more detail on the platform, refer to the 2018 DHP study out of InsideTracker [20]). User data was collected between January 2012 and September 2023. The DHP described in this study is not intended to function as a medical device as defined by the FDA's guidelines on Software as a Medical Device [43]. Rather, it is designed to deliver personalized, evidence-based recommendations aimed at supporting general wellness and health optimization through behavior change. Users are informed at the time of consent to DHP use that the product does not provide clinical diagnoses or treatment plans, nor does it replace the role of licensed healthcare professionals. This distinction is consistent with

regulatory frameworks, which allow for wellness-oriented digital health tools that encourage healthy living, provided they do not claim to diagnose or treat specific medical conditions. The DHP is fully HIPAA compliant and SOC2 Type II certified, and all user data is encrypted, obfuscated, anonymized and de-identified. Additionally, users are directed to consult a medical professional in cases where they receive biomarker results that are in a range that would indicate emergent clinical risk.

### Recruitment *of* participants

Users between 18 and 85 residing in North America were recruited through company marketing. In accordance with the company's terms and conditions, privacy policy, and product and research agreements, users consented to their data being used in blinded, aggregated research analyses. User retention is managed through email, SMS communication, and webinars.The BRANY Institutional Review Board (IRB) determined this work was exempt from review. Guidelines for observational research in tissue samples from human subjects were followed. This study employed data from 20,342 participants that met our analysis inclusion requirement; the presence of two or more blood tests at least 90 days apart.

### Biomarker collection and analysis

Blood samples were collected and analyzed by Clinical Laboratory Improvement Amendments (CLIA)–approved, third-party clinical labs. Participants were instructed to fast for 12 hours prior to the phlebotomy. Results were uploaded to the platform via electronic integration. Participant sample size per biomarker is not uniform.

### Biomarker dataset preparation

In our dataset, occasional outlier values were observed that were deemed implausible (e.g., fasting glucose < 65 mg/dL; daily steps > 60,000/day). To remove anomalous outliers in a systematic way, we used the Interquartile Range (IQR) method of identifying outliers, removing data points which fell below Q1–2 IQR or above Q3 + 2 IQR. Average activity/sleep tracker data for 90 days prior to the blood test was considered paired with that blood test.

### Calculation *of* Polygenic Risk Scores (PGS)

Genomic variants (SNPs) comprising PGSs were derived from publicly available GWAS summary statistics (https://www.ebi.ac.uk/gwas, https://www.pgscatalog.org, see S5 Table for PGS identifiers). Scores were calculated by summing the product of effect allele doses weighted by the beta coefficient for each SNP. Variant p-value thresholds were chosen based on optimization of respective PGS-blood biomarker correlation in the entire user cohort with both blood and genomics datasets (~1000–1500 depending on the blood biomarker at the time of analysis). Genotyping data was derived from a combination of a custom InsideTracker array and third-party arrays such as 23andMe and Ancestry. Not all variants for any particular PGS were genotyped on every array; proxies for missing SNPs were extracted via the "LDlinkR" package using the Utah Residents (CEPH) with Northern and Western European ancestry (CEU) population (R2 > 0.8 cut-off). Risk categories were determined as the top 10% (T3), middle 80% (T2), and lowest 10% (T1) of continuous scores across the population, and associations between PGS strata and biomarker changes were evaluated using analysis of variance.. These thresholds allow high and low risk to be defined as the top and bottom tertiles of the population, a division shown in studies of genetic risk to have clinical relevance across multiple conditions [44].

### Longitudinal analysis

Longitudinal changes between baseline and follow-up measurements were assessed using a two-tailed Wilcoxon signed-rank test for paired samples. For subsequent analysis, baseline data were filtered to include only those participants whose baseline value was outside the optimal range for each biomarker. Optimal range is defined as an ideal range that lies

within the laboratory's designated clinically normal reference range and is determined by a combination of one or more of the following criteria: age, sex, and ethnicity and established based on prior knowledge, data-driven assessment of healthy participant variation in the NHANES cohort, or both. A prior knowledge determination would include evaluation of research studies that evaluate the impact of different biomarker levels on mortality or disease risk. Where possible, special consideration was taken to include age and ethnicity (but not ancestry at this time, see discussion). Parameters for determination of a healthy individual in the NHANES cohort include: no evidence of cardiovascular disease, hypertension, diabetes or cancer, BMI between 18 and 25, and non-smokers. Full documentation of the derivation of optimal zones for each marker can be found in S11 Table. Users were informed what zone their biomarkers fell into: Low, normal-low, optimal, normal-high, and high. Since suboptimal values can be too high or too low, we separately evaluated users with baseline levels above or below optimal. Only biomarkers with at least 200 participants out-of-range at baseline were included.

For long-term longitudinal analysis, a Wilcoxon signed-rank test was performed between samples paired between baseline and each subsequent test, as well as between the second blood test and each subsequent test, to assess the significance of improvement at both first and subsequent follow-ups to the baseline blood levels. Significance was considered as $p <= 0.05$.

Regression-to-the-mean (RTM) effects were quantified separately for each biomarker using parameters estimated from a baseline reference population. For each biomarker, we calculated the population mean at baseline, denoted $\mu$, and the test–retest correlation between draws 1 and 2, denoted $\rho_{1,2}$. The expected RTM-attributable change for an individual with baseline value $x_1$ was estimated as

$$\Delta RTM = (1 - \rho_{1,2})(\mu - x_1).$$

The cohort-level RTM contribution was then obtained by averaging $\Delta RTM$ across all individuals in the longitudinal analysis cohort.

## Responder characteristics analysis

Users with high baseline TC were selected from the longitudinal cohort for the responder characteristics analysis based on the availability of wearable fitness tracker data from 30 days prior to their baseline blood draw, as well as from during the at least 90 days between baseline and follow-up draw. Data for deep, REM, and total sleep, step count, and vO2 max was compared between users who improved TC levels by follow-up and those who did not, by two-tailed Welch's t-test, at baseline. Users were considered "responders" if their biomarker levels changed enough to move into a preferable clinical range or zone (range definitions described above) or if their biomarker level changed in a preferential direction by 10% or more. The latter threshold allowed users who presented at baseline with extremely suboptimal biomarker levels to be included as responders even if their improvement did not constitute a zone shift; for cholesterol in particular, a 10% decrease is associated with clinically meaningful health improvements [45]. Average changes from baseline to the time between blood tests were also compared between improvement groups by t-test to identify significant differences. Due to the smaller sample size and noisiness of wearable data, significance was considered as $p <= 0.1$, to reduce the likelihood of Type II errors. Results should be considered exploratory in nature.

## Engagement *with* action plan feature and improvement analysis: categorical "zone-shift" outcomes

To evaluate whether creating an Action Plan (AP) that contained at least one recommendation targeted at addressing a specific suboptimal biomarker was associated with categorical improvements in biomarker zones between baseline (BL) and follow-up (FU), zone-shift analyses were conducted separately for each biomarker and BL stratum ("Biomarker High at BL" and "Biomarker Low at BL") as follows. For each biomarker, we defined the following cohorts for contrast: participants who created an AP that included at least one recommendation targeting a given biomarker during their baseline

to follow-up blood draw interval, and participants with no AP of any kind created during their interval. Participants who created APs for other biomarkers but not the focal biomarker were excluded from this contrast. Participants with missing BL or FU zone values were excluded from the zone-shift analyses for that biomarker. As described above, biomarkers are categorized into standardized zones indicating whether they should be targeted for health improvements and the direction of improvement for optimal health. A successful outcome was defined as a user improving ≥1 zone, i.e., their FU zone ranked closer to optimal than their BL zone by at least one step for a given biomarker. For each biomarker×outcome×cohort contrast, we formed a 2×2 contingency table (rows = cohorts; columns = outcome: Yes/No). Pearson's chi-squared test without continuity correction was performed using the Rstats library (stats::chisq.test).

## Supporting information

**S1 Table. Blood biomarker sample size and summary statistics.**
(PDF)

**S2 Table. Physiological biomarker & PGS sample size and summary statistics.**
(PDF)

**S3 Table. Users with higher than optimal baseline biomarker levels: baseline vs. follow-up.**
(PDF)

**S4 Table. Users with lower than optimal baseline biomarker levels: baseline vs. follow-up.**
(PDF)

**S5 Table. Correlation between polygenic traits and baseline blood biomarkers.**
(PDF)

**S6 Table. Correlation between polygenic traits and change in blood biomarker levels between baseline and follow-up test.**
(PDF)

**S7 Table. Baseline and post-baseline personal fitness tracker data compared between users who improved high cholesterol levels compared to non-improved users.**
(PDF)

**S8 Table. Zone-transition outcomes by cohort;** *Action Plan targeting biomarker vs no Action Plan; biomarkers above optimal at baseline*.
(PDF)

**S9 Table. Zone-transition outcomes by cohort;** *Action Plan targeting biomarker vs no Action Plan; biomarkers below optimal at baseline*.
(PDF)

**S10 Table. Quantification of Regression-to-the-Mean Effects by Biomarker.**
(PDF)

**S11 Table. Definitions and Sources of Biomarker "Optimal" Ranges.**
(PDF)

**S1 Fig. Biomarker cross-correlation among DHP users.**
(PDF)

**S2 Fig. Action Plan adoption by DHP users.**
(PDF)

**S3 Fig. Long-term trajectories in biomarkers optimal at baseline (negative-exposure comparator).**
(PDF)

**S4 Fig. Age vs. biomarker level (mean ± SE) in 5-year bins.**
(PDF)

**S1 Data. Analysis dataset.** Contains raw data used in analysis (demographic, blood biomarker, and wearables-derived data).
(CSV)

**S1 Methods. Supplementary methods.**
(PDF)

## Author contributions

**Conceptualization:** Nimisha Schneider, Gil Blander, Renee Deehan.

**Data curation:** Nimisha Schneider, Paul Fabian.

**Formal analysis:** Nimisha Schneider, Paul Fabian.

**Investigation:** Nimisha Schneider, Renee Deehan.

**Methodology:** Nimisha Schneider, Michelle Cawley, Bartek Nogal, Renee Deehan.

**Project administration:** Nimisha Schneider, Renee Deehan.

**Supervision:** Nimisha Schneider, Gil Blander, Renee Deehan.

**Validation:** Paul Fabian.

**Visualization:** Nimisha Schneider.

**Writing – original draft:** Nimisha Schneider, Renee Deehan.

**Writing – review & editing:** Nimisha Schneider, Michelle Cawley, Bartek Nogal, Gil Blander, Renee Deehan.

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
