## [Decision Letter · Decision Letter 0]

17 Jun 2025

Response to Reviewers
Revised Manuscript with Track Changes
Manuscript
**Journal Requirements:**
**Additional Editor Comments (if provided):**
**Reviewers' Comments:**

**Comments to the Author**

1. Does this manuscript meet PLOS Digital Health’s publication criteria?

Reviewer #1: Yes

Reviewer #2: Partly

Reviewer #3: No

2. Has the statistical analysis been performed appropriately and rigorously?

Reviewer #1: Yes

Reviewer #2: Yes

Reviewer #3: I don't know

3. Have the authors made all data underlying the findings in their manuscript fully available (please refer to the Data Availability Statement at the start of the manuscript PDF file)?

Reviewer #1: Yes

Reviewer #2: No

Reviewer #3: No

4. Is the manuscript presented in an intelligible fashion and written in standard English?

Reviewer #1: Yes

Reviewer #2: Yes

Reviewer #3: Yes

Reviewer #1: The manuscript is well written, clearly structured, and addresses a relevant and timely topic in the field of digital health. While the manuscript is generally well written, the objective of the study should be stated more explicitly, both in the abstract and in the main text.

This long-term study demonstrates the potential of personalized digital health platforms, analyzing biomarkers, genetics, and activity data to foster sustained improvements in health. The paper makes a valuable contribution to ongoing efforts to leverage passive data collection for physiological insights, and it will likely be of interest to researchers, clinicians, and engineers working for for understanding and improving health span.

Additionally, some tables lack clear legends or explanations for abbreviations used, and the quality of the figures is relatively low, making it difficult to interpret key details. Also, the authors should address the need to integrate these tools with professional guidance, mitigating risks and ensuring ethical use.

Reviewer #2: This study makes a meaningful contribution to digital health literature by demonstrating real-world, sustained improvements in biomarkers through a digital platform. The inclusion of polygenic risk and wearable data adds an important nuance.

Strengths of the manuscript:

(a) Strong integration of multiple data modalities over a long follow-up period.

(b) Clear evidence that biomarker improvements are possible and sustainable in a real-world setting.

(c) Sound statistical handling of observational data.

(d) A well-written and accessible manuscript with relevant findings for the field of personalized digital health.

However, there are some limitations need attention:

(a) Equity and inclusion - The cohort lacks diversity in geography, race, and socioeconomic status. Addressing how these results could translate to underrepresented populations is essential, especially for global health relevance.

(b) Intervention clarity - The recommendations given to users are not sufficiently described. Were they standardized, personalized manually, or algorithmically generated? What adherence data exist?

(c) Ethical scope - While consent and IRB exemption are noted, more detail on ethical management of genetic and wearable data would strengthen trust.

(d) Causality caveat - The study is observational and self-selected, which the authors acknowledge. Still, clearer articulation of how future randomized studies could validate these associations would be helpful.

(e) Software transparency - The DHP algorithm is proprietary. Providing more transparency or simulated examples of how recommendations are derived would support open science goals.

Further details below as per PLOS Digitial Health Criteria:

1. The manuscript addresses a highly relevant and broad interest topic in digital health - longitudinal tracking of biomarkers and personalized interventions. The study integrates blood biomarkers, polygenic risk scores, and wearable fitness data over a multi-year period in a real-world setting. This combination is novel and forward-thinking in digital health research. However, the study’s generalizability is limited due to its demographically homogenous and self-selected population (predominantly White, male, tech-savvy, and U.S.-based), which may not reflect global digital health contexts, especially in the Global South. Methodological rigor is strong, but more context-specific implications and inclusivity are needed.

2. The study is methodologically sound for a retrospective observational design. The manuscript uses appropriate statistical tests for longitudinal data, including Wilcoxon signed-rank and Mann-Whitney U tests. The study has a large sample size (over 20,000 users), long-term data, and multiple types of evidence that support the conclusions about the potential benefits of personalized digital health platforms. Effect sizes are reported alongside p-values, and genetic correlations are examined through regression models. However, the lack of a control group limits causal inference. Addressing regression to the mean strengthens credibility, though some methodological choices like thresholds for improvement warrant additional justification.

3. A de-identified minimal dataset is included, which is commendable. However, the genetic data central to some key findings are not publicly available due to consent restrictions. This limits full reproducibility. The authors should clarify how external researchers might gain access under ethical safeguards.

4. The writing is clear, well-organized, and mostly free from grammatical issues. Figures and tables are appropriately used. The tone is accessible for non-specialists, although some terms like “optimal zone” could be better explained for clarity across global audiences.

Overall recommendation - Major Revision; especially accounting for the equity concern, (otherwise, I would caveat this with a minor revision)

Major revision should clarify:

(i) The cohort is overwhelmingly White, male, and U.S.-based. While this may reflect the user base of the commercial platform, it limits the generalizability and relevance of the findings to the broader global digital health community particularly populations in low-resource or underserved settings.

(ii) The manuscript should include a thoughtful discussion of how the results could (or could not) apply to more diverse populations. This includes barriers such as cost, digital access, and cultural applicability. Discuss the practical accessibility of the platform for broader communities, especially in the Global South. Is the service affordable or adaptable for use in public health systems? If not, how could the insights still be useful?

(iii) Consider including a statement on plans for future studies or platform adaptations that aim to improve inclusivity, if any.

Minor revisions should clarify:

(i) The mechanics and adherence of the personalized interventions - while the proprietary nature of the platform’s recommendation engine is acknowledged, but more transparency is needed. A simplified explanation or example of how personalized suggestions are generated would help satisfy open science expectations. Clarify how recommendations were delivered and whether user adherence was measured or assumed + some parts of the methods, such as how users were engaged and retained.

(ii) While the minimal dataset is appreciated, limitations around access to genetic data should be explained more clearly, including how other researchers might access it under appropriate safeguards. A bit more transparency on the DHP algorithm logic, even in simplified form.

Reviewer #3: This study examines blood biomarkers and fitness tracker data in users of a digital health platform. Unfortunately, there’s no clear research question. Reading it through, I understand the authors are comparing health parameters from the same individuals longitudinally, and attributing positive changes to the use of the platform, and the lack thereof to genetic factors. However, the manuscript does not define primary and secondary outcome measures (what is most important in the available data?), and a research question is not formulated. Examining a dataset to make sense of it can be used to formulate hypothesis, but is not original research in itself.

**Do you want your identity to be public for this peer review?** For information about this choice, including consent withdrawal, please see our Privacy Policy

Reviewer #1: No

Reviewer #2: No

Reviewer #3: No

**Figure resubmission:****Reproducibility:** To enhance the reproducibility of your results, we recommend that authors of applicable studies deposit laboratory protocols in protocols.io, where a protocol can be assigned its own identifier (DOI) such that it can be cited independently in the future. Additionally, PLOS ONE offers an option to publish peer-reviewed clinical study protocols. Read more information on sharing protocols at https://plos.org/protocols?utm_medium=editorial-email&utm_source=authorletters&utm_campaign=protocols

---

## [Editor Report · Decision Letter 1]

23 Sep 2025

Response to Reviewers
Revised Manuscript with Track Changes
Manuscript
**Journal Requirements:**
**Additional Editor Comments:**

I have several questions/suggestions,

1) The levels of compliance among the users may be a key issue. How about the compliance of the users? How did you assess the level of user compliance? Did the platform have any specific way to improve compliance/cope with low compliance?

2) Causal inference is a major concern of the reviewers. For causal inference, Mendelian randomization may be used: you have polygenic risk scores (PGS) as instrumental variables, BMI/LDL-c/… as exposure variables, the changes in the values as outcome variables, and the levels of compliance as disturbance variables (if the levels of compliance were not the same for all users). This is also for what you mentioned in lines 78-79: to assess whether genetic risk influenced the degree of improvement.

https://pmc.ncbi.nlm.nih.gov/articles/PMC7849343/

https://mr-dictionary.mrcieu.ac.uk/term/prs/

https://mr-dictionary.mrcieu.ac.uk/term/prs-approach/

?>**Reviewers' Comments:****Figure resubmission:**

**Reproducibility:**To enhance the reproducibility of your results, we recommend that authors of applicable studies deposit laboratory protocols in protocols.io, where a protocol can be assigned its own identifier (DOI) such that it can be cited independently in the future. Additionally, PLOS ONE offers an option to publish peer-reviewed clinical study protocols. Read more information on sharing protocols at https://plos.org/protocols?utm_medium=editorial-email&utm_source=authorletters&utm_campaign=protocols

---

## [Decision Letter · Decision Letter 2]

7 Dec 2025

Response to Reviewers
Revised Manuscript with Track Changes
Manuscript
**Journal Requirements:**

**Additional Editor Comments (if provided):**

Lines 36-37:

users who increased their daily steps by ~1,000 and those with higher REM sleep percentages were more likely to shift cholesterol in a healthier direction.

—However, Figure 5A shows that at some time points, the difference between the two groups is far greater than 1,000 steps？

Lines 164-169：

A high proportion of users were able to improve HgbA1c, fasting glucose, triglycerides, and hsCRP levels (79.3%, 74.2%, 76.2%, and 70.7% respectively; Figure 1E, C, B, F), but relatively few could completely optimize them (21.4%, 31.1%, 29.6%, and 25.4%, respectively, of suboptimal users improved).

— What are the differences between these two groups (improve vs. completely optimize)—behavioral differences, differences in Polygenic Risk Scores, or others?

LDL-c also showed this pattern: a relatively higher proportion (20.4%) of users able to improve vs (5.3%) optimize their initially high levels by their follow-up blood test.

— What are the differences between these two groups (improve vs. completely optimize)—behavioral differences, differences in Polygenic Risk Scores, or others?

Lines 190-194:

The ferritin PGS most significantly associated with longitudinal changes with the corresponding serum biomarker (Figure 4C): high baseline ferritin levels decreased on average upon retest in T1, T2, and T3 users; however, the decrease was significantly larger for users in T1 than those in T2 or T3.

  ?>**Reviewers' Comments:**

**Comments to the Author**

Reviewer #1: All comments have been addressed

Reviewer #2: (No Response)

publication criteria?

Reviewer #1: Yes

Reviewer #2: Partly

3. Has the statistical analysis been performed appropriately and rigorously?

Reviewer #1: Yes

Reviewer #2: Yes

4. Have the authors made all data underlying the findings in their manuscript fully available (please refer to the Data Availability Statement at the start of the manuscript PDF file)?

Reviewer #1: Yes

Reviewer #2: No

5. Is the manuscript presented in an intelligible fashion and written in standard English?

Reviewer #1: Yes

Reviewer #2: Yes

Reviewer #1: I believe the authors have incorporated the suggested recommendations, which has significantly enhanced the article's robustness and ensured its compliance with the journal's standards.

Reviewer #2: The authors have made substantial and good-faith improvements across both R1 and R2. Most methodological, transparency, and framing concerns are now resolved or adequately acknowledged. The remaining issues are minor clarifications rather than structural or analytic weaknesses. The manuscript is now publishable with targeted refinements, particularly concerning reproducibility of “optimal zones,” explicit caution around exploratory sub-analyses, and fully tempering causal language.

How well prior concerns were addressed:

Addressed sufficiently;

- Clarified cohort selection and inclusion criteria.

- Improved description of biomarker zones and optimal ranges.

- Added explicit explanation of retrospective design and non-causal framework.

- Strengthened explanation of polygenic risk score derivation and interpretation.

- Expanded limitations regarding selection bias, demographic skew, lack of medical history, and device heterogeneity.

- Clarified the handling of wearables data and rationale for the relaxed significance threshold.

Addressed partially;

- Regression-to-the-mean concerns are acknowledged but could be quantified or tested (e.g., via sensitivity analysis on biomarker variance at baseline).

- “Optimal zone” definition remains partly proprietary; could be expanded for transparency.

- Claims around healthspan or preventive potential still occasionally drift toward interpretive overreach—ensure language remains observational.

Remaining suggestions (minor);

1. The “optimal zone” definition, although better described, still mixes clinical reference ranges with proprietary definitions; greater clarity would strengthen reproducibility. Provide a supplementary table listing the rationale and sources for each biomarker’s “optimal” band, distinguishing clinical vs internal definitions.

2. Clarify whether the “minimal dataset” includes all variables used for statistical testing (e.g., wearables-derived REM%, step counts, biomarker zone assignments).

3. The small n for the wearable-sleep subgroup (n=126) and relaxed p-value threshold (0.1) should be highlighted more prominently as exploratory.

4. The interpretation of sustained improvements across >5 draws still risks regression-to-the-mean inflation; the authors acknowledge this, but could further quantify the likely magnitude of this effect

5. Ensure all language aligns with associational findings (e.g., “correlates with” rather than “impacts” or “leads to”).

6. Code transparency (if feasible) - Include code snippets or analytic scripts used for the Wilcoxon analyses, PGS stratification, and outlier detection, even if only pseudocode.

7. Add one or two sentence highlighting limitations of ancestry mismatch (CEU proxies) and potential bias for non-European users.

**Do you want your identity to be public for this peer review?** For information about this choice, including consent withdrawal, please see our Privacy Policy

Reviewer #1: None

Reviewer #2: No

**Figure resubmission:**

**Reproducibility:** To enhance the reproducibility of your results, we recommend that authors of applicable studies deposit laboratory protocols in protocols.io, where a protocol can be assigned its own identifier (DOI) such that it can be cited independently in the future. Additionally, PLOS ONE offers an option to publish peer-reviewed clinical study protocols. Read more information on sharing protocols at https://plos.org/protocols?utm_medium=editorial-email&utm_source=authorletters&utm_campaign=protocols

---

## [Editor Report · Decision Letter 3]

11 Feb 2026

Improvements in blood and fitness tracker biomarkers in a longitudinal real-world cohort of digital health platform users

PDIG-D-25-00206R3

Dear Schneider,

We are pleased to inform you that your manuscript 'Improvements in blood and fitness tracker biomarkers in a longitudinal real-world cohort of digital health platform users' has been provisionally accepted for publication in PLOS Digital Health.

Best regards,

Pengxu Wei

Academic Editor

PLOS Digital Health